# Investigating sensory response to physical discomfort in children with autism spectrum disorder using near-infrared spectroscopy

**Larissa C. Schudlo**[1,2], **Evdokia Anagnostou**[1,3], **Tom Chau**[1,4], **Krissy Doyle-Thomas**[1,5]*

**1** Bloorview Research Institute, Holland Bloorview Kids Rehabilitation Hospital, Toronto, Canada,
**2** Computer and Biomedical Engineering Department, Ryerson University, Toronto, Canada, **3** Faculty of Medicine, Institute of Medical Science, University of Toronto, Toronto, Canada, **4** Institute of Biomaterials and Biomedical Engineering, University of Toronto, Toronto, Canada, **5** School of Health and Community Services, Mohawk College, Hamilton, Canada

* krissy.doyle-thomas@mohawkcollege.ca

**Data Availability Statement:** Demographic data underlying the results presented in the study are available from (Brain-CODE https://www.braincode.ca/). NIRS data and pain intensity ratings cannot be

## Abstract

Self-reporting of pain can be difficult in populations with communication challenges or atypical sensory processing, such as children with autism spectrum disorder (ASD). Consequently, pain can go untreated. An objective method to identify discomfort would be valuable to individuals unable to express or recognize their own bodily distress. Near-infrared spectroscopy (NIRS) is a brain-imaging modality that is suited for this application. We evaluated the potential of detecting a cortical response to discomfort in the ASD population using NIRS. Using a continuous-wave spectrometer, prefrontal and parietal measures were collected from 15 males with ASD and 7 typically developing (TD) males 10–15 years of age. Participants were exposed to a noxious cold stimulus by immersing their hands in cold water and tepid water as a baseline task. Across all participants, the magnitude and timing of the cold and tepid water-induced brain responses were significantly different ($p < 0.001$). The effect of the task on the brain response depended on the study group (group x task: $p < 0.001$), with the ASD group exhibiting a blunted response to the cold stimulus. Findings suggest that NIRS may serve as a tool for objective pain assessment and atypical sensory processing.

## Introduction

Recognizing pain or discomfort is an assessment typically made by oneself or a caregiver. This assessment can be challenging for individuals with atypical sensory processing and/or have difficulty understanding or recognizing their distress. An example population that faces these challenges are children with Autism Spectrum Disorder (ASD). Pain management can be especially difficult for caregivers and clinicians of these individuals. Unaddressed pain can delay injury treatment, increase the risk of further injury or cause psychological distress leading to self-injurious or aggressive behavior [1]. An objective means of identifying and evaluating pain would be of value for individuals unable to recognize and/or express their bodily distress.

shared publicly because participant consent is currently unavailable for a third party transfer. Data are available from the Holland Bloorview Research Ethics Board Chair (Alison Williams via awilliams@hollandbloorview.ca) for researchers who meet the criteria for access to confidential data.

**Funding:** This study was financially supported by the Centre for Leadership at Holland Bloorview Kids Rehabilitation Hospital (LS, EA, TC, KD), the Natural Science and Engineering Research Council (NSERC) Research Tools & Instrumentation Grant (EQPEQ421950-12) (TC) and the Lillian and Don Wright Foundation (TC).

**Competing interests:** The authors have declared that no competing interests exist.

## Autism spectrum disorder (ASD)

ASD is a heterogeneous group of neurodevelopmental disorders characterized by social communication and behavioral challenges [2]. The core symptoms of ASD are often accompanied by sensory processing challenges [3–5]. Manifestation of sensory processing challenges varies substantially across the population, with respect both depth and breadth [6]. Majority of individuals with ASD report hypo- or hyper- sensitivities in multiple sensory domains (*e.g.* sound, touch, pain) [5,7]. Aberrant pain sensitivity is one of the most prevalent sensory processing abnormalities [3]. Some individuals with ASD are intolerant of seemingly unremarkable painful stimuli, while others can be completely unresponsive to pain [8].

## Assessment of physical pain or discomfort

Physical pain or discomfort can manifest as a variety of behavioral changes such as frowning, fidgeting, or increased irritability or aggression [1]. While a caregiver might use these indicators to recognize that an individual is in pain, they are non-specific. Physiological measurements can be more objective and definitive [9].

The basis of aberrant sensory processing in developmental disorders, including ASD, remains largely unknown. However, Sensory Processing Disorder and an atypical response to ordinary stimuli is believed to stem from brain rather than peripheral nervous system dysfunction [10,11]. Specifically, atypical sensory modulation is believed to stem from the brain's inability to appropriately regulate the received sensory information to produce a suitable output [5]. Thus, directly measuring the brain, the epicenter of the sensory processing pathway, may provide the most effective indicator of one's state of discomfort than peripheral measurements.

## Functional brain imaging of sensory processing and pain

Pain is a complex, multi-dimensional integration of sensory, affective and cognitive components. These three dimensions interact to modulate the painful experience and activate a broad network of brain regions. For acute pain, these regions include the somatosensory cortex, medial and dorsolateral prefrontal cortex, and the inferior and posterior parietal cortex [12–15]. The prefrontal and parietal cortices are primarily involved in the affective and cognitive aspects of pain, while the somatosensory cortex is involved in the sensory-discriminative dimension [16].

Studies have shown that functional brain activity due to sensory processing may differ between individuals with ASD and typically developing individuals for tactile [17], auditory [18] and visual stimuli [18], despite similar perceptual reports [17]. However, these studies have primarily involved adult participants, and imaging of the pediatric brain has been limited [19]. This is due to the ethical considerations and challenges in imaging younger individuals [20–22] as well as the constraint of using modalities such as functional Magnetic Resonance Imaging (fMRI) and Positron Emission Tomography (PET). These modalities require an individual to remain still, in a supine position, and in a loud environment for an extended period of time. These demands make imaging children and populations with sensory sensitivities highly challenging. Thus, our understanding of the neural pain signature in children with ASD is limited to date [19]. However, neurological processing and pain tolerance can vary with brain development and age [21,23], warranting explicit study of specific populations and/or age ranges. As an alternative to fMRI, near-infrared spectroscopy (NIRS) may be better suited for clinical assessments of the pediatric and/or ASD population.

### Near-infrared spectroscopy (NIRS)

NIRS is a non-invasive optical imaging modality that measures hemodynamic activity in the outer layers of the cerebral cortex. The individual dons a cap containing a number of near-infrared light sources and detectors to acquire brain measurements. It is portable, soundless and can accommodate subject movement and does not require any paste or gel for setup. As such, it can be used under a wider breadth of experimental paradigms and in naturalistic settings (*e.g.*, in clinic, therapy, or social settings).

For individuals with ASD, NIRS has been used to evaluate functional brain activity under various executive function [24–26] and emotional-based [27,28] paradigms. Although this modality has been used to assess cortical activity in response to noxious stimuli in the typically developing population [29–34] its use to identify pain or discomfort, in children with ASD has not yet been explored.

### Cold pressor task

Cutaneous stimulation is most commonly used to experimentally induce pain or discomfort [12]. The cold pressor task (CPT) has been used most used in the pediatric population [35]. With the CPT, an individual immerses their hand in cold water for as long as tolerable. This innocuous stimulus temporarily induces a slowly-mounting pain response from which the individual can voluntarily withdraw. The experience is similar to that of naturally occurring pain [36]. Previous work with both fMRI [37] and NIRS [33] has demonstrated that the CPT can induce a significant brain response in typically developing individuals. Given these considerations, the CPT is well suited to assess children with ASD.

### Objectives

In this work, we investigated the feasibility of characterizing cortical response to a noxious sensory stimulus in children with ASD using NIRS. Specifically, our primary objectives were to determine, on a group-level:

i. If a significant cortical response to a noxious cold stimulus could be identified using NIRS in children;

ii. If the cortical response to a noxious cold stimulus differed significantly between children with ASD and typically developing (TD) children in amplitude and/or timing.

Accurately identifying a state of pain or discomfort through cortical activity alone would provide an objective means of identifying pain or discomfort in children with ASD without verbal report. Additionally, the ability to identify such response using NIRS could provide a practical alternative to MR imaging. This could permit consideration of a wider variety of functional assessments and more diverse populations, accelerating our understanding and treatment of pain and aberrant sensory processing in ASD.

## Methods

### Participants

Twenty-two male participants were recruited for this study through Holland Bloorview Kids Rehabilitation Hospital (Toronto, Ontario, Canada) and the Province of Ontario Developmental (POND) Network. Participants in the POND Network who had agreed to receive study recruitment emails received a study flyer via email and were invited to contact the study's research coordinator for more information if interested. Fifteen participants were individuals

with ASD, and 7 were typically-developing (TD) controls. All participants were between the ages of 10–15 years and met the following inclusion/exclusion criteria: normal or corrected to normal vision, able to understand and communicate in English, no history of chronic pain, hypertension or heart conditions, and no hand abrasions.

Participants with ASD had a clinical diagnosis of the disorder. The majority of clinical diagnoses were confirmed using the Autism Diagnostic Observational Schedule-2 (ADOS-2) or the Autism Diagnostic Observational Schedule (ADOS), and the Autism Diagnostic Interview–Revised (ADI-R) (see Table 3 for detailed information). TD participants had no diagnosis of any developmental or psychiatric disorders or conditions, no history of neurological disease or family history of ASD and were medication-free at the time of the study.

Ethics approval was obtained from Holland Bloorview Kids Rehabilitation Hospital, and informed written consent was obtained from each participant prior to study participation. Participants were given a description of the study and their understanding of the study was assessed through a series of questions. It was also confirmed that participants met the inclusion/exclusion criterion during the consent process and were eligible to participate. Upon successfully demonstrating their understanding of the study and confirming eligibly criterion were met, and if they were still willing to participate, participants and their parent or guardian signed the consent form. As a token of appreciation for their participation, participants were given a gift card (regardless of successful study completion). A parents or guardian was present during the consent process, but not during the experimental protocol.

## Experimental protocol

Each participant attended one experimental session, approximately 2 hours long. Initial set-up up and execution of the experimental protocol took approximately 30min and 45min, respectively. The remainder of the time was used to obtain written consent and administer pre- and post-protocol questionnaires.

**Experimental set-up.** Participants sat alongside 3 water-filled containers facing a computer screen and wearing the necessary data collection sensors (Fig 1). Two of the containers were filled with tepid, room temperature water (between 23–26˚C), and the third container was filled with cool water (maintained at 10˚C ± 1˚C, as recommended by von Baeyer *et al.* [21]). The middle container (*i.e.* the acclimation container) always contained tepid water. One of the end containers (either the front or back) also contained tepid water (*i.e.*, the control container), and the other container contained cool water (*i.e.*, the stimulus container). The locations of the control and stimulus containers (front/back) were randomized across participants. Water temperature was automatically monitored and maintained using a thermometer-controlled electric pump. In each container, a pump gently circulated the water to prevent any warming around the hand and arm.

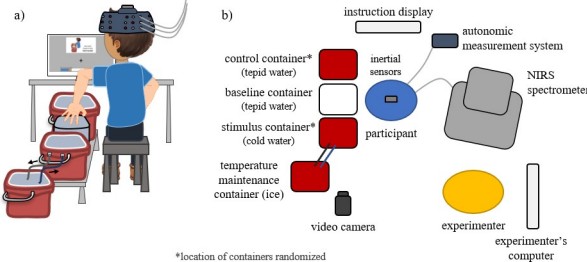

**Fig 1.** Experimental setup a) view from behind the participant and b) schematic of instrumentation from above.

**Experimental timing.** The timing paradigm of the session is shown in Fig 2. At the start of the protocol, a 5 min period of baseline rest was collected. Participants were asked to relax, remain as still as possible and look at a fixation cross on the computer interface. Following this baseline period, they were visually cued to immerse their hand in the acclimation (middle) container. After a 2 min acclimation period, participants were visually cued to move their hand into either the front or back container.

When placing their hand in the stimulus (cold water) container, participants were instructed to keep their hand in the water for as long as comfortably tolerable or until a maximum of 3 minutes was reached. When ready, participants moved their hand back to the middle (acclimation) container to allow the functional response to the cold stimulus to subside. Following a 2min acclimation interval, participants were visually cued again to move their hand into either the front or back container. Each time participants placed their hand in the control container, they were visually cued to move their hand back to the acclimation container after approximately 1 minute.

Participant moved their hand in and out of the control container to measure functional brain activity under similar motor and decision-making demands as the CPT task, in the absence of sensory stimulation. Thus, a comparison of measurements during the CPT and the control intervals should reflect the response to the cold stimulus, while the effects of variables such as movement or decision-making processes ought to be suppressed.

Movement from the middle container to either the front or back container was repeated 6 times during the experimental session (3 per container). During the experiment, the timing of hand movement was recorded by the experimenter via software input. Timing was later confirmed using a video recording of the participant's hand that was synchronized with measurements on the NIR spectrometer.

## Functional measurements

NIRS measurements were acquired using the ETG-4000 continuous-wave Optical Topography System from Hitachi Medical Co. (Japan). Measurements of cerebral oxygenation were acquired from 46 locations over the prefrontal and parietal cortices (Fig 3). We chose to monitor these brain regions to capture components of the cortical response reflective of pain or discomfort, minimizing the response solely due to sensory stimulation. Furthermore, the scalp-to-cortex distance in these brain regions is small enough to capture cortical activity via NIRS, whereas the scalp is thicker and less likely to capture cortical activity in the somatosensory cortex [38].

The 18 NIR light emitters and 15 photodetectors were secured to the participant's head using a custom-made head cap that was adjusted to place the channels over specific locations of the brain (according to the 10–20 International System). Measurements channels situated midway between emitter-detector pairs separated by 3cm were collected for analysis. Each NIR light emitter emitted NIR light at wavelengths of 695nm and 830nm. The optical signals were sampled at 10Hz.

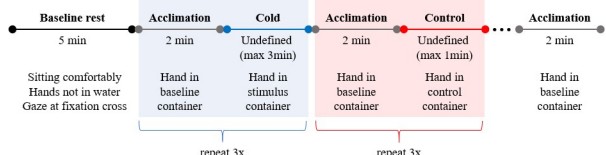

**Fig 2. Timing of the experimental paradigm.** Note that the order of the cold and control conditions were randomized.

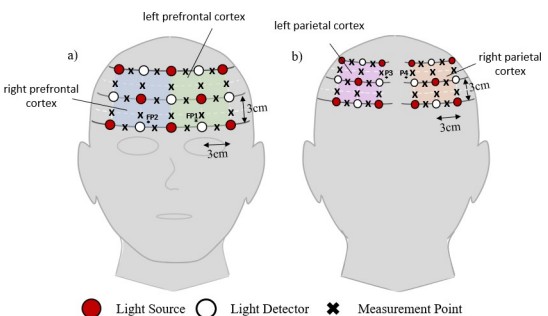

**Fig 3. Source-detector configuration.** Three grids of NIR emitters and photodetectors were placed on the participant's head over the a) prefrontal and b) right and left parietal cortices. Each shaded circle (18) represents the location of an emitter in the configuration, and each unshaded circle (15) represents a photo detector. Each 'x' represents a measurement channel. Locations in the 10–20 International System are demarcated with an '*'.

In addition to cortical activity, blood volume pulse was measured by securing an optical sensor to the index finger of the participant's right hand (ProComp Infinity System, Thought Tech Technologies, Montreal, Canada). To assess movement, a wireless accelerometer and gyroscope were secured to the top of participant's head via the NIRS head cap.

## Behavioral measurements

Age and handedness were collected from each participant. Full scale IQ was obtained through the POND Network and was determined using the Wechsler Abbreviated Scale of Intelligence (WASI)–II, Standford-Binet Intelligence Score, or the Wechsler Intelligence Scale for Children (WISC)–V.

The duration of time immersed in the cold water was used as an objective measure of pain tolerance. At the end of the session, participants were also asked to assess their pain level by rating the intensity of pain experienced for both cold and tepid water on a numeric scale from 0–10.

## NIRS data analysis

### Preprocessing

Note that we will use 'task interval' to collectively refer to acclimation, stimulus, or control intervals (in Fig 2). Prior to analysis, the data were visually inspected to remove any channels or task intervals that were highly contaminated with movement artifacts or low signal quality. Additionally, data from 5 participants (all from the ASD group) were removed prior to analysis due to technical issues with the data collection equipment (P207, P213) or substantial movement-related artifacts (P202, P204, P205) that affected the majority of the measurement channels.

Although NIRS provides both oxygenated and deoxygenated hemoglobin concentrations, only oxygenated hemoglobin concentration ([HbO]) measurements were considered in this analysis as it yielded a more prominent change in response to the stimulus. [HbO] concentration. Signals were low-pass filtered using a 3$^{rd}$ order Type II Chebyshev filter with a passband cut off frequency of 0.1Hz, and stopband cut off frequency of 0.5Hz. This filter suppressed high frequency noise stemming from physiological phenomena such as cardiac activity (between 0.8–1.2Hz), low-frequency artifacts due to respiration (approximately 0.3Hz), and arterial pulsations (*i.e.* the Mayer wave, approximately 0.1Hz) [39].

## Feature extraction

The hemodynamic response occurs 5-8s following the onset of a stimulus. To eliminate any brain activation or movement artifacts due to hand movement in/out of the water-filled containers, the first 10s of each task interval were removed. The remaining data (*i.e.*, 10s onward) were considered for analysis.

The amplitude of [HbO] measurements for each NIRS measurement channel for each task interval was normalized by subtracting the initial value from the measured signal. Four brain regions were considered for analysis: left and right prefrontal cortices and left and right parietal cortices (Fig 3). For each of the 4 brain regions, the average of all regional measurements channels was determined. Averaging signals is a common signal processing technique to minimize noise and capture the true underlying response. From these average responses, the maximum [HbO] was calculated. This maximum [HbO] value was used in statistical analyses.

Because participants dictated the duration of the cold stimulus trials, the length of the trials varied across participants and task repetition. All control trials were 60s long, while the stimulus trials were up to 180s. To compare the evolution of the hemodynamic response between trial types, trial duration ought to be consistent. Thus, two different comparisons of brain activity were considered for statistical analysis of functional brain measurements: i) maximum [HbO] across the full task intervals and ii) maximum [HbO] within the first 60s of the task intervals.

## Statistical analysis

Demographic variables (age and IQ) were compared across the two groups (ASD *vs* TD) using a t-test. Subjective pain ratings, pain/discomfort tolerance, maximum [HbO] amplitude, and time to peak [HbO] amplitude values were compared using mixed linear model analyses. In these analyses, group, task type, brain region, and/or trial number were modelled as fixed effects, and subject was modelled as a random effect. Variables considered for each analysis are listed in Table 1. A *p*-value of 0.05 was set as the threshold for statistical significance, and was Bonferroni-corrected for multiple comparisons.

## Results

Four participants in the ASD group (P204, P205, P207 and P208) had difficulty performing the cold-water portion of the protocol, reporting that the water was too cold. These participants received verbal coaching from the experimenter, who encouraged the participants to keep their hand in the water for at least 10 seconds. Participants were still free to remove their hand from the water whenever they chose. Two participants were able to complete the protocol with verbal encouragement (P205 and P208), while the other two participants were not (P204,

**Table 1. Variables considered for statistical analysis of magnitude and timing of functional brain response.**

|  |  | Magnitude of brain response | Timing of brain response | Pain Tolerance | Pain Ratings |
|---|---|---|---|---|---|
|  | **Variable** | Levels | | | |
| **Fixed Effects** | Group | ASD, TD | ASD, TD | ASD, TD | ASD, TD |
|  | Tsk Type | Stimulus (cold), Control (tepid) | Stimulus (cold), Control (tepid) | Stimulus (cold) | Stimulus (cold), Control (tepid) |
|  | Brain Region | Left PFC, Right PFC, Left Parietal, Right Parietal | Left PFC, Right PFC, Left Parietal, Right Parietal |  |  |
|  | Trial Number | 1, 2, 3 | 1, 2, 3 | 1, 2, 3 | 1, 2, 3 |
|  | Response | Maximum [HbO] | Time to maximum [HbO] | Trial duration | Pain rating |

P207). Thus, functional brain measurements from these two participants were not included in the analysis. Otherwise, no participants reported any distress or agitation induced by the experimental paradigm. NIRS measurements from 3 other participants in the ASD groups were also removed from analysis: two due to significant movement artifacts (P202, 213), and one due to technical issues with the measurement equipment (P205). Results are presented for the remaining 10 participants in the ASD group.

## Participant characteristics

Average demographic characteristics within and across study groups are shown in Table 2. Diagnostic information for ASD participants is shows in Table 3. The two participant groups were not significantly different with respect to age ($p = 0.76$, $t_{16} = -0.309$), but differed significantly with respect to IQ ($p = 0.01$, $t_{15} = -3.23$).

## Subjective pain ratings

Average self-assessment pain scores are shown in Table 2 for all participants that were able to complete the assessment. Note that 2 individuals from the ASD group were unable to complete the self-assessment (P203, P205). The pain ratings were significantly different between the two task types ($p < 0.01$, $F_{70} = 241.663$) but did not different significantly between the two study groups ($p = 0.628$, $F_{14} = 0.245$).

## Pain/Discomfort tolerance

Fig 4 shows the duration of time each participant was able to hold their hand in cold water for each trial. On average, the stimulus (cold water) and control (tepid water) trials were $166.30 \pm 41.5s$ and $64.32 \pm 2.8s$ in duration, respectively, for the TD group, and $94.56 \pm 74.7s$ and $61.82 \pm 6.8s$ long, respectively, for the ASD group (Table 1). Statistical analysis of the cold-water trial durations showed no significant differences were observed between the two groups or across the three trials (group: $p = 0.09$, $F_{45} = 7.483$; trial: $p = 0.406$, $F_{31} = 0.929$).

## Brain response–amplitude and timing of cortical response

A comparison of brain activity across the two types of tasks are show in Fig 5, where Fig 5A shows a comparison of maximum [HbO] across the full task interval, and Fig 5B shows a comparison of the maximum [HbO] within the first 60s of the task intervals.

**Table 2. Participant characteristics, average self-assessment of pain ratings and pain tolerance.**

| Demographic information | | All (N = 17) | TD (N = 7) | ASD (N = 10) | p-value |
|---|---|---|---|---|---|
| | Age (years) | 12.5 ± 2.0 | 12.9± 1.8 | 12.6 ± 2.2 | age: $p = 0.76$ |
| | Handedness | 1L, 16 R | 0L, 7R | 1L, 9R | |
| | Full Scale IQ | 97.3 ± 20.5 | 113.2 ± 12.9* | 87.8 ± 18.3 | IQ: $p = 0.01$ |
| Subjective Pain Ratings | | All (N = 16) | TD (N = 7) | ASD (N = 9) | |
| | Tepid water | 0.21 ± 0.6 | 0.33 ± 0.9 | 0.11 ± 0.3 | groups: $p = 0.628$ |
| | Cold water | 5.88 ± 2.9 | 6.17 ± 1.8 | 5.65 + 3.5 | task: $p < 0.01$ |
| Pain tolerance (duration of trial) | | All (N = 17) | TD (N = 7) | ASD (N = 10) | |
| | Tepid water (s) | 62.68 ± 5.9 | 64.32 ± 2.8 | 61.58 ± 7.3 | groups: $p = 0.09$ |
| | Cold water (s) | 127.38 ± 68.9 | 166.30 ± 41.5 | 108.8 ± 73.7 | trials: $p = 0.406$ |

Age and IQ values show (mean ± standard deviation). Handedness shows number of left-handed (L) and right handed (R) participants. *IQ was not available for one participant (P107) in the TD group. Values represent mean ± standard deviation.

**Table 3. Diagnostic information for ASD participants.**

| ADOS-2 | Module | No. Participants | Social Affect (SA) | Restricted and Repetitive Behaviour (RRB) | |
|---|---|---|---|---|---|
| | 3 | N = 9 | 9.9 ± 3.9 | 5.4 ± 2.1 | |
| ADOS | Module | No. Participants | Communication | Stereotyped Behaviours + Restricted Interests | Reciprocal Social Interaction |
| | 4 | N = 1 | 6 | 5 | 9 |
| | 3 | N = 3 | 2.3 ± 0.56 | 1.3 ± 0.58 | 6 ± 2 |
| ADI-R | Module | No. Participants | Communication | Restricted, Repetitive, and Stereotyped Patterns | Reciprocal Social Interaction |
| | -- | N = 13 | 17.4 ± 4.6 | 4 ± 0.9 | 23 ± 4.9 |

ADOS-2, ADOS, and ADI-R sub-scale values show (mean ± standard deviation). ADOS was not available for two participants (P205 and P212) and ADI-R was not available for two participants (P202 and P210).

To compare differences in sensory responsiveness across the two study groups, the full evolution of the brain response induced by the cold, noxious stimulus was considered. The brain response (*i.e.* peak relative [HbO]) was significantly affected by the type of task performed (cold vs tepid water, $p < 0.001$, $F_{1,249} = 159.5$). The effect of each task on the brain also depended significantly on the study group considered (group x task interaction: $p < 0.001$, $F_{1,249} = 32.4$). As seen in Fig 5B, the difference in cortical response to the cold and control stimuli is large in TD group than the ASD group. Additionally, the brain response was significantly affected by the brain region monitored ($p = 0.003$, $F_{3, 245} = 4.7$), with post-hoc pair-wise comparisons revealing a significant difference between the right prefrontal and the left parietal cortices only.

Similar statistical findings surfaced when considering the brain response within the first 60s of the task intervals. The type of task performed significantly affected the brain response ($p < 0.001$, $F_{1, 257} = 36.1$), with the effect of each task depending on the study group (group x task interaction effect: $p = 0.049$, $F_{1, 275} = 3.9$). The brain response was also significantly

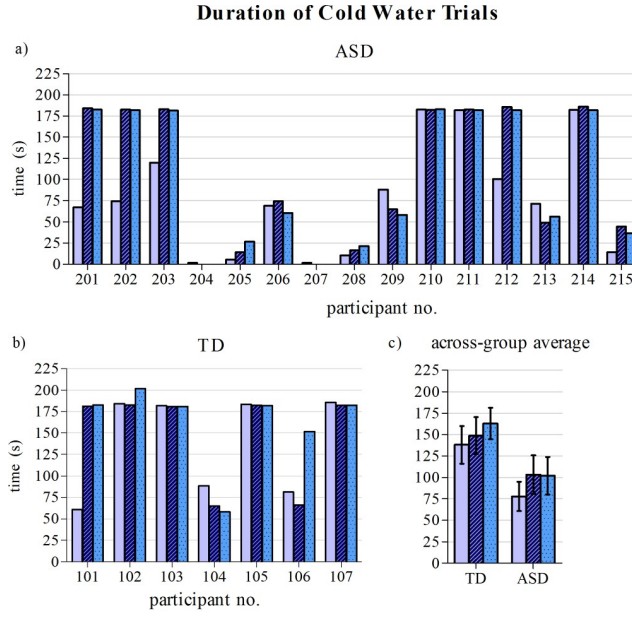

**Fig 4.** Duration of each of the three cold-water trials for a) TD participants b) participants with ASD and c) across-group averages. Error bars represent standard error of the mean. * indicates participant excluded from analysis of functional brain data.

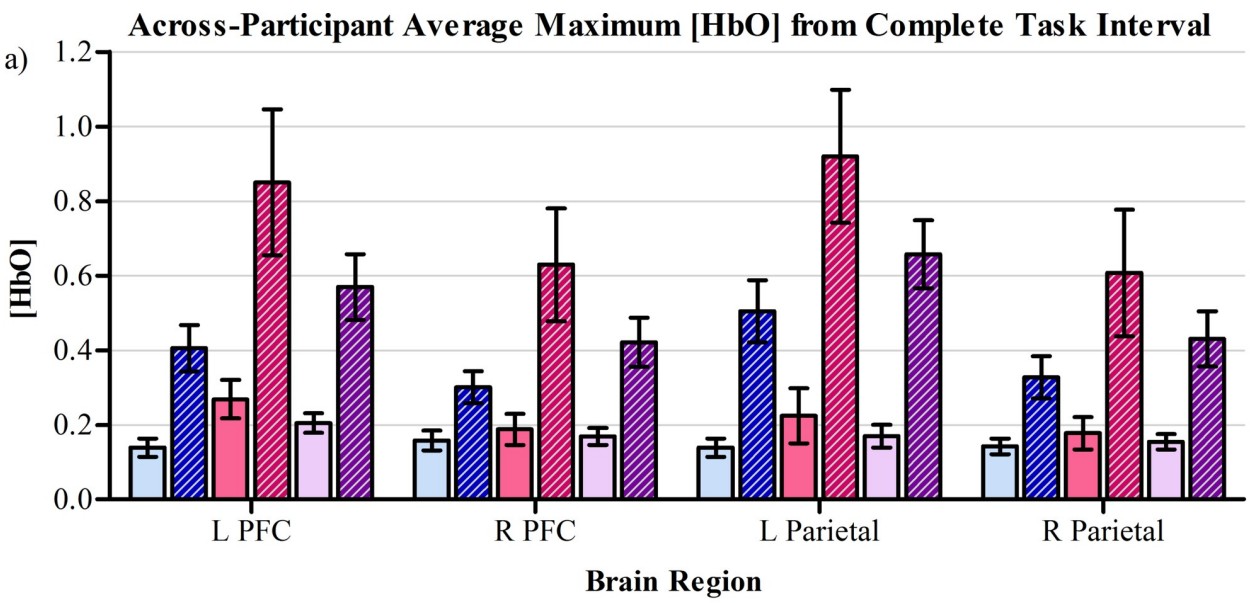

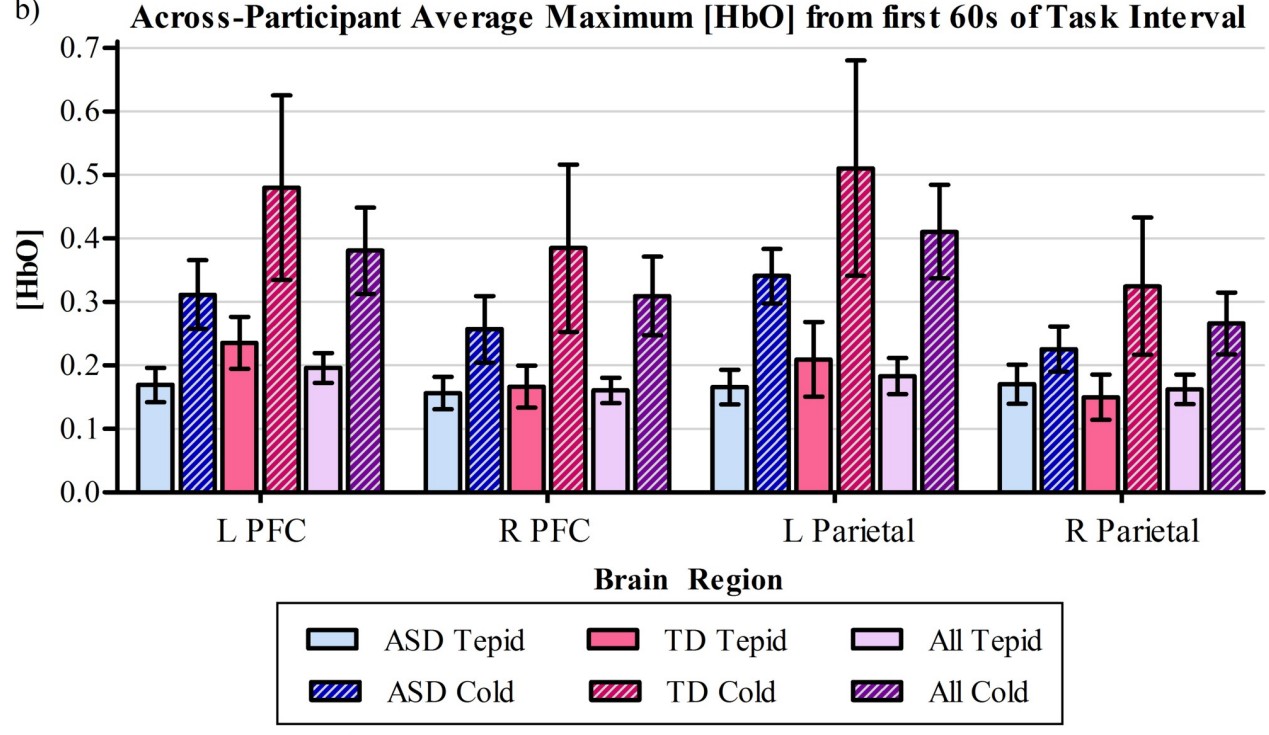

error bars represent standard error of the mean

**Fig 5.** Across-participant average functional brain response a) from the full task trials and b) from the first 60s of the task trials. The peak relative [HbO] of the cold-water or tepid-water trials were average for each of the cold-water and tepid-water trials for each of the 4 brain regions, for each study group. Error bars represent standard error of the mean.

different across trials ($p < 0.001$, $F_{2, 275} = 10.0$), with a significant difference between trials 1 and 2, and 1 and 3.

The timing of the brain response (*i.e.* time to peak [HbO]) was significantly different between the two tasks (cold *vs* tepid water: $p < 0.001$, $F_{1,250} = 338.5$). The effect of task on the

timing of the brain response also depended on the group and trial number (task x group interaction: $p < 0.001$, $F_{1, 250} = 23.6$, group x task x interval interaction: $p < 0.001$, $F_{2, 247} = 9.9$).

## Discussion

There remains an unmet need of establishing a reliable and clinically feasible method to objectively detect a neural pain signature or study the atypical sensory processing in clinical populations. Most pain studies have been performed on adults, due, in part, to the challenges of conducting traditional functional brain imaging with children [20]. In this study, we investigated the potential of using NIRS to detect a response to a noxious cold stimulus in the children with ASD using the CPT. To our knowledge, this is the first study to investigate the utility of NIRS for detecting cortical changes during sensory pain processing in the pediatric ASD population.

### Detecting a cortical response to pain

Relative to the tepid water control stimulus, a significantly different brain response evoked by the cold water (in terms of magnitude and timing) in both the prefrontal and parietal cortices across both study groups (Fig 5A). The subjective pain ratings align with these findings (Table 1), indicating that participants did indeed experience unpleasantness from the cold water. Given the overall concordance between subjective pain ratings and the peak oxy-hemoglobin concentrations, the heightened brain activity was most likely discomfort related.

Pain is a subjective experience that is modulated not only by stimulus intensity, but also biological and psychological factors, such as emotions, distractions and one's attitude [20,40]. These factors may have contributed to the varying degrees of brain activation and the differences in pain/discomfort tolerance observed across participants. The magnitude of an evoked brain response is also proportional to the strength of the evoking stimulus [31]. The water temperature of the CPT is known to affect the perceived level of pain [23]. A more intense stimulus may yield more pronounced changes in brain activity. Future studies may consider varying the stimulus intensity to explore differential pain thresholds across individuals and the sensitivity of NIRS to gradations of induced pain/discomfort.

### Aberrant sensory processing in the ASD population

Individuals with atypical sensory processing often exhibit distorted responses to a perturbation on the cortical level [41]. In accordance, our analysis also revealed that the brain response to the different stimuli was significantly influenced by neurodevelopmental status. As seen in Fig 6, the difference in response elicited by the cold-water stimulus is more pronounced and sustained in the TD group than in the ASD group across all 4 brain regions. In contrast, the response to the innocuous, tepid water stimulus was similar between the two groups (not shown in Fig 6). A similar finding of a blunted response to heat was observed by Failla et al. (2018) in an fMRI study with the adult participants with ASD [42].

Altered patterns of sensory responsiveness is a ubiquitous characteristic of ASD [6], with individuals presenting with unusually high or low thresholds to sensory stimuli. This is believed to stem from a disruption in the brain's ability to communicate simultaneously synthesize all the presented information [10,43]. However, the mechanisms of how this processing is altered in ASD are not well understood at present [6].

The brain is not a passive recipient of sensory stimuli. One's perception of pain does not always accurately reflect the stimulus intensity or injury severity [15]. Our findings show that, despite similarities in subjective pain ratings (Table 2), functional brain activity in response to the noxious sensory stimulus was different between ASD and TD individuals. This discrepancy

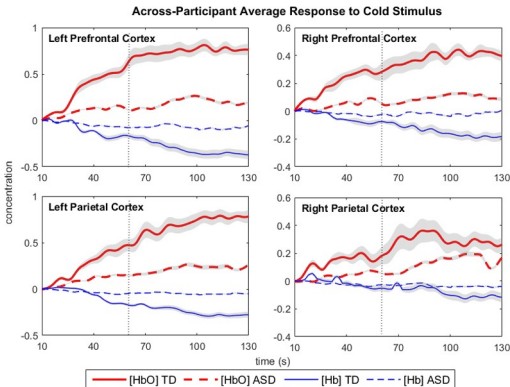

**Fig 6. Average across-participant brain response to noxious cold stimulus for each study group in the left prefrontal, right prefrontal, left parietal and right parietal cortices.** Average oxygenated and deoxygenated hemoglobin ([HbO] and [Hb], respectively) responses are plotted with shaded areas representing standard error the mean for each time point. Dashed vertical lines indicate the 60s mark of the trial.

is consistent with previous fMRI work [17]. This highlights the need for an objective measure of pain assessment for the ASD population, that does not rely on self-evaluations of affective constructs. Given the differences in response, population-specific models of pain perception are likely necessary to establish a clinically viable pain assessment tool. Individuals with sensory processing disorders can have altered reactions to sensory stimuli that depend on the type of stimulus, its timing, and the number of sensory systems stimulated at once [11]. Thus, a larger heterogenous sample of the population and different types of stimuli ought to be considered in future work to evaluate generalizability of these findings. Capturing the heterogeneity in ASD would yield greater understanding of the neurobiological underpinnings of atypical sensory processing and pain perception.

## Brain regions implicated in pain processing

Our findings show that differences in brain response evoked by the noxious and innocuous stimuli varied with the brain region monitored. An interaction between task type and brain region was observed between the left and right hemispheres of the parietal cortex. The average response was smaller in the right parietal cortex than in the left, primarily due to the lateralized response to the cold-water stimulus (Fig 5).

Interestingly, due to the right hemisphere's role in attentional systems [44,45], the cortical response to pain or discomfort is often lateralized more dominantly to the right hemisphere, regardless of the side of stimulation [46]. Since participants placed their *left* hands in cold water, it would be expected the *right*, contralateral hemisphere would exhibit a larger response if the response was primarily sensory. Yet, we observed a diminished response in the right hemisphere. The left hemisphere of the brain tends to be responsible for perceiving the sensory component of a painful stimulus, whereas the right hemisphere is responsible for the affective component [47]. Given the familiarity of the stimulus employed, it is not surprising that the cortical response could be more influenced by sensory rather than affective component of pain perception, yielding a left-lateralized.

## Effect of task repetition

Repeated delivery of noxious stimulus can lead to a decline in perceived intensity. Areas of the brain that attenuate pain become more responsive, while region that respond to pain become

less responsive [48]. Our results show that the difference in brain response evoked by the noxious and control stimuli was influenced by the number of task repetitions when considering a 60s task duration. A large number of participants were able to withstand the cold water for the maximum permitted duration by the third trial (Fig 4). As participant repeated the cold-water trials, they likely acclimatized to the stimulus. Although a significant response to the cold stimulus was maintained across all trials, the effect of varying the stimulus intensity on a per-subject or per-trial basis ought to be explored in future work.

## Comparison to previous functional imaging work

Functional imaging studies using fMRI and PET have provided consistent evidence that painful, thermal stimuli elicit distinct response patterns in the cerebral cortex [12–15,20,37,49]. However, both heightened and decreased levels of blood flow being observed [14,46]. This is, in part, due to variations in stimulus characteristics, including type, intensity, duration, and site of application. fMRI studies focused specifically on pain processing in the ASD population have documented a diminished neural response to sensory stimulation in individuals with ASD relative to control groups [17,42], consistent with our findings.

A limited number of studies evaluated the cortical response to thermal stimuli using NIRS [29–33], all of which considered measurements from the prefrontal and/or sensory cortices in the typically developing/developed population. Ours is the first study to consider a large measurement area from both the prefrontal and parietal regions to investigate pain-induced hemodynamic activity and evaluate activity amongst these disjoint regions involved in pain/sensory processing. Of previous NIRS studies investigating pain-induced activity, only Barati *et al.* (2013) employed the CPT to induce discomfort. In this work, a significant task-induced increase in cortical activity in 4 prefrontal measurements was found, while reported pain scores decreased across the 3 repetitions of the task [33]. Although this study was conducted with typically developed adult participants, the findings of Barati *et al.* resonate with our observations of elevated hemodynamic activity in response to the CPT. Heightened hemodynamic activity in response to painful stimuli was also found in other works utilizing NIRS [29,30,32].

## Limitations

**ASD population representation.**   Understanding the biological underpinning of pain processing in ASD is complicated by the heterogeneity of the disorder. Individuals who exhibit aberrant sensory processing, specifically aberrant sensory modulation, can exhibit hypo- or hyper- sensitivity to stimuli. In this study, four participants struggled to perform the task because of the cold-water temperature, all of which were in the ASD group (P204, P205, P207 and P208) (Fig 4). Of these four, two found the cold water entirely unbearable and were unable to complete the task (P204 and P207).

It is likely that the portion of the ASD population with hyposensitivity to pain is more represented in this study than those who exhibits hypersensitivity. Individuals with hyposensitivity stand to benefit more from an objective means of detecting pain, as it could be used to alert themselves or caregivers of unrecognized distress. However, understanding the mechanisms of sensory processing in the ASD population requires study of a larger breadth of the population. Individuals with hypersensitivity to pain are less likely to adhere to the study protocols, or even volunteer to participate. One solution would be to accommodate each individual's pain tolerance by making subject-specific adjustments to the stimulus intensity in real-time. A comparison of neurological response at each individual's stimulation threshold may better highlight functional brain difference between ASD and TD populations or, on the contrary, highlight differences in pain tolerability when cortical response is similar.

The four participants who struggled to perform the CPT on their first attempt received moderate verbal encouragement from the experimenter thereafter. This proved effective for two of the participants (P205 and P208). Coping mechanisms and distractors can increase pain tolerance in children [50]. However, the effect of such strategies on the mechanisms of brain function is not well understood, especially in the ASD population. Further study in this area may better elucidate the mechanisms of pain processing. Incorporating verbal coaching into our protocol may also increase our ability to recruit and evaluate individuals with hyper-sensitivity.

It should be noted that medications can potentially affect hemodynamic and/or cortical activity, as well as pain perception and attentional abilities. While it is common for individuals with ASD to take medications for, for example, ADHD, that can affect prefrontal activity, this was not controlled in this study. The study sample size is not large enough to draw any meaningful conclusions regarding the effect of medications on the cortical responses. Further study should take this into consideration, especially for developing models of pain detection for real-world, clinical use.

It should also be noted that sensory processing disorders can also present in the general population. Participants should be screened explicitly for sensory processing disorders and severity in future work, and considerations of this subgroup ought to be considered.

**Sample size.** This preliminary study was conducted with a small number of participants and study groups were imbalanced. The sample size challenges the generalizability of the results. It also makes it more challenging to detect significant statistical effects with a smaller sample [51]. Yet, a notable difference in cortical response to the noxious and non-noxious stimuli was detected in addition to a significant interaction between stimulus type and study group. The fact that significant differences were observed given our sample size suggests that the observed effects are indeed real. These preliminary results indicate further investigation with a larger sample size is warranted.

Because pain tolerance [52] and ASD symptomatology [53] can vary with sex, only considered male participants. However, extended study involving female participant is necessary. Similarly, a large span of ages should be considered in future studies. Study groups were also not matched for IQ in this work. Although this a potential confounding factor, it has been found that cognitive level (IQ) or overall developmental level is not related to abnormal sensory reactivity in children with ASD [3].

**Clinical relevance of findings and expanded applications.** The pain or discomfort experience during the CPT is a close analog to naturally occurring pain [36]. The cortical response to acute physical pain evaluated in this work would translate to scenarios where the onset of pain occurred during the acquisition of functional measurements. For example, imagine an individual undergoing a dental procedure who was unable to communicate the occurrence of a painful event. Consideration of different types of sensory/painful stimuli, and varied intensity levels, would also be necessary to enhance the generalizability of findings. Additionally, consideration of other autonomic measurements, such as heart rate, vagal tone, and electrodermal activity, in conjunction with cortical activity may better capture the physiological response to sensory stimuli [54,55]. Our focus was on assessing the utility of NIRS to measure a response. However, autonomic measurements have shown to be reliable indicators of hyper or hypo sensitivity and help separate physiological pain from perception. A multi-modal analysis could be considered in future work.

Objectively identifying and quantifying pain would also be of value to other populations with communication challenges such as individuals with dementia, traumatic brain injury, or individuals who are post-surgery. NIRS-based investigation in other populations could be considered.

Lastly, this work evaluated average, group-level difference in brain response to noxious cold stimulus. A clinically viable tool to detect and assess pain would require classification of a single response to the stimulus using machine-intelligent algorithm. A modality such as NIRS permits the necessary quantity and diversity of data to be collected to establish such an algorithm, and ought to be considered in future work.

## Conclusion

Currently, no practical and reliable method to objectively detect and evaluate pediatric pain exists. This study represents a first step in using NIRS as a clinical tool to characterize the neurological response to a noxious stimulus in ASD without the need for subjective evaluation. Our preliminary results support the use of NIRS as a viable modality for this application. We observed that hemodynamic measurements taken with NIRS in the prefrontal and parietal cortices exhibited a significant change in response (magnitude and timing) to noxious cold stimulus, with notable differences between the ASD and TD study groups.

## Acknowledgments

The authors would like the thank Ka Lun Tam for his technical work in instrumentation development and Stephanie Chow for performing the data collection. The authors would also like to acknowledge Cathy Petta, Pam Green, and Paul Smith for their advice during project ideation.

## Author Contributions

**Conceptualization:** Larissa C. Schudlo, Evdokia Anagnostou, Krissy Doyle-Thomas.

**Data curation:** Larissa C. Schudlo, Krissy Doyle-Thomas.

**Formal analysis:** Larissa C. Schudlo, Krissy Doyle-Thomas.

**Funding acquisition:** Evdokia Anagnostou, Tom Chau, Krissy Doyle-Thomas.

**Investigation:** Larissa C. Schudlo, Evdokia Anagnostou, Tom Chau, Krissy Doyle-Thomas.

**Methodology:** Larissa C. Schudlo, Evdokia Anagnostou, Tom Chau, Krissy Doyle-Thomas.

**Project administration:** Larissa C. Schudlo, Krissy Doyle-Thomas.

**Resources:** Evdokia Anagnostou, Tom Chau, Krissy Doyle-Thomas.

**Software:** Larissa C. Schudlo.

**Supervision:** Larissa C. Schudlo, Krissy Doyle-Thomas.

**Writing – original draft:** Larissa C. Schudlo.

**Writing – review & editing:** Larissa C. Schudlo, Tom Chau, Krissy Doyle-Thomas.

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
