## [Decision Letter · Decision Letter 0]

30 Mar 2021

PONE-D-21-06009

Investigating Sensory Response to Physical Discomfort in Children with Autism Spectrum Disorder using Near-Infrared Spectroscopy

PLOS ONE

Dear Dr. Doyle-Thomas,

Thank you for submitting your manuscript to PLOS ONE. After careful consideration, we feel that it has merit but does not fully meet PLOS ONE’s publication criteria as it currently stands. Therefore, we invite you to submit a revised version of the manuscript that addresses the points raised during the review process.

Two experts in the field have carefully reviewed the manuscript entitled, "Investigating Sensory Response to Physical Discomfort in Children with Autism Spectrum Disorder using Near-Infrared Spectroscopy". Their comments are appended below.

The first reviewer acknowledged the manuscript is considerably well written pointing out some important suggestions which will be strengthen the manuscript.

The reviewer #2 is also well evaluated the manuscript, however, this reviewer raised several critical issues to be considered before publication. 

Judging from these favorable comments, I would like to be revised the manuscript according to each critique.

We look forward to receiving your revised manuscript.

Kind regards,

Manabu Sakakibara, Ph.D.

Academic Editor

PLOS ONE

Journal Requirements:

3. You indicated that you had ethical approval for your study. In your Methods section, please ensure you have also stated whether you obtained consent from parents or guardians of the minors included in the study or whether the research ethics committee or IRB specifically waived the need for their consent.

Reviewers' comments:

Reviewer's Responses to Questions

**Comments to the Author**

1. Is the manuscript technically sound, and do the data support the conclusions?

Reviewer #1: Partly

Reviewer #2: Yes

2. Has the statistical analysis been performed appropriately and rigorously? 

Reviewer #1: Yes

Reviewer #2: Yes

3. Have the authors made all data underlying the findings in their manuscript fully available?

Reviewer #1: No

Reviewer #2: Yes

4. Is the manuscript presented in an intelligible fashion and written in standard English?

Reviewer #1: Yes

Reviewer #2: Yes

5. Review Comments to the Author

Reviewer #1: Well written study with valid clinical foundation. The use of the cold challenge is established as safe with children and used in pain research. Findings are important to the development of this area of research.

Limitations:

1) As noted by the authors, the effects of medication on these children were not controlled for or reported. This may have a significant impact of the perception of pain and the attentional abilities of the child.

2) The authors site the work of Dunn, Schoen, and Miller demonstrating an understanding of sensory processing disorder and it's classifications. There is no description of the sensory processing disorder diagnosis or severity in this manuscript. Children with over- and under-responsive types of SPD will naturally respond differently to stimuli.

3) Since children with SPD may occur in the general population, often without co-morbid diagnoses, control group children should have been screened for SPD as well.

Suggestions to consider in future studies:

1) Children with SPD have altered reactions to sensory habituation based on the type of stimuli and the presentation timing of the stimuli. Single modal stimulation often has a muted response, while multi-modal stimuli has a delayed response that is typically additive across trials. See the Sensory Challenge Protocol studies from Schaaf and Miller.

2) Another consideration would be the effect of decreased vagal tone on the physiologic response to pain. Typically children with SPD demonstrate lower heart rate at rest. When in pain, most individuals will show an increase in HR due to parasympathetic responses to noxious stimuli. By measuring HR while looking at the blood flow to the brain, there may be a way to determine physiologic response vs. perception of pain.

3) Electrodermal measurements have also been shown to be a reliable indicator of sensory over- or under-arousal status.

Overall, I think this is a good article. I would add data on the SPD sub-type with which the children present, or list it as a limitation of this study.

Reviewer #2: PONE-D-21-06009_reviewer

Investigating Sensory Response to Physical Discomfort in Children with Autism Spectrum Disorder using Near-Infrared Spectroscopy

Recommendation: Major revision

In the present manuscript the authors report on their findings of male children with autism spectrum disorder (ASD) their response to painful stimulus and the utility of near infrared spectroscopy (NIRS) to identify focal changes in brain oxygenation, correlating with a cortical response to the painful stimulus. Overall the authors observed a blunted reaction to the painful stimulus in patients with ASD, relative to typically developing children.

This study was well performed and the resulting manuscript is overall well written, with a well researched and lengthy discussion section. The conclusions appear well supported by the data and are certainly relevant to the scientific community. However, several critiques should be considered prior to publication.

1. While well written, the manuscript is too long. Specifically, the introduction and discussion should be abbreviated to include only the most pertinent information. Most of the background information should be made more concise. In general, any claim should be supported by a relevant citation. The first several paragraphs of the introduction have many claims without citation and are quite verbose. Recommend making the introduction much more concise.

2. A full discussion of the subject inclusion and exclusion is recommended. This should include both demographic inclusion criteria as well as clinical. Furthermore, the recruitment strategy should be discussed. Also, if any compensation was involved. Please describe the consenting process and other ethical considerations, including the parental roles.

3. This reviewer questions the decision to include the “excluded” subjects in the analysis. Including these subjects, and also analyzing the cohort without them leads to a confusing description of the results. Furthermore, table 2 is quite confusion with these two groups described (with and without usable NIR data). Recommend only reporting results on subjects with a complete dataset.

4. Table 2 should also include p values.

5. The results section should report on the data only. Further description of methods should be placed in the “methods” section, including the section starting on page 15 line 354

6. PLOS authors have the option to publish the peer review history of their article (what does this mean?). If published, this will include your full peer review and any attached files.

Reviewer #1: No

Reviewer #2: **Yes: **Kurt Yaeger

---

## [Author Response · Author response to Decision Letter 0]

6 May 2021

Reviewer #1

1. As noted by the authors, the effects of medication on these children were not controlled for or reported. This may have a significant impact of the perception of pain and the attentional abilities of the child.

>> We have added these potential effects to our discussion regarding medication.

Added text:

"Limitations

ASD Population

It should be noted that medications can potentially affect hemodynamic and/or cortical activity, as well as pain perception and attentional abilities."

2. The authors site the work of Dunn, Schoen, and Miller demonstrating an understanding of sensory processing disorder and its classifications. There is no description of the sensory processing disorder diagnosis or severity in this manuscript. Children with over- and under-responsive types of SPD will naturally respond differently to stimuli.

>> Unfortunately, we do not have this information specifically regarding sensory processing diagnoses or severity for the participants of this study. This has been added as a limitation and consideration for future work. 

Added text:

"Limitations

ASD population representation

It should also be noted that sensory processing disorders can also present in the general population. Participants should be screened explicitly for sensory processing disorders and severity in future work, and considerations of this subgroup ought to be considered." 

3. Since children with SPD may occur in the general population, often without co-morbid diagnoses, control group children should have been screened for SPD as well.

Suggestions to consider in future studies

>> This information has been added to the manuscript.

Added text:

"Limitations

ASD population representation

It should also be noted that sensory processing disorders can also in the general population. Participants should be screened explicitly for sensory processing disorders and severity in future work, and considerations of this subgroup ought to be considered." 

4. Children with SPD have altered reactions to sensory habituation based on the type of stimuli and the presentation timing of the stimuli. Single modal stimulation often has a muted response, while multi-modal stimuli has a delayed response that is typically additive across trials. See the Sensory Challenge Protocol studies from Schaaf and Miller.

>> We have included this information and reference to the work of Schaaf and Miller in the manuscript. 

Added text:

"Assessment of physical pain or discomfort 

However, Sensory Processing Disorder and an atypical response to ordinary stimuli is believed to stem from brain rather than peripheral nervous system dysfunction (10,11). Specifically, atypical sensory modulation is believed to stem from the brain’s inability to appropriately regulate the received sensory information to produce a suitable output (5).

Aberrant sensory processing in the ASD population

Individuals with sensory processing disorders can have altered reactions to sensory stimuli that depend on the type of stimulus, its timing, and the number of sensory systems stimulated at once (11). Thus, a larger heterogenous sample of the population and different types of stimuli ought to be considered in future work to evaluate generalizability of these findings. Capturing the heterogeneity in ASD would yield greater understanding of the neurobiological underpinnings of atypical sensory processing and pain perception."

5. Another consideration would be the effect of decreased vagal tone on the physiologic response to pain. Typically children with SPD demonstrate lower heart rate at rest. When in pain, most individuals will show an increase in HR due to parasympathetic responses to noxious stimuli. By measuring HR while looking at the blood flow to the brain, there may be a way to determine physiologic response vs. perception of pain.

>>Consideration of vagal tone to evaluate sensory processing in our paradigm has been added to the manuscript. 

Added text

"Clinical relevance of findings and expanded applications

Additionally, consideration of other autonomic measurements, such as heart rate, vagal tone, and electrodermal activity, in conjunction with cortical activity may better capture the physiological response to sensory stimuli (53,54). Our focus was on assessing the utility of NIRS to measure a response. However, autonomic measurements have shown to be reliable indicators of hyper or hypo sensitivity and help separate physiological pain from perception. A multi-modal analysis could be considered in future work."

6. Electrodermal measurements have also been shown to be a reliable indicator of sensory over- or under-arousal status.

>>Consideration of electrodermal activity to evaluate sensory processing in our paradigm has been added to the manuscript. 

Added text

"Clinical relevance of findings and expanded applications

Additionally, consideration of other autonomic measurements, such as heart rate, vagal tone, and electrodermal activity, in conjunction with cortical activity may better capture the physiological response to sensory stimuli (53,54). Our focus was on assessing the utility of NIRS to measure a response. However, autonomic measurements have shown to be reliable indicators of hyper or hypo sensitivity and help separate physiological pain from perception. A multi-modal analysis could be considered in future work."

Overall, I think this is a good article. I would add data on the SPD sub-type with which the children present, or list it as a limitation of this study.

Reviewer #2

1. While well written, the manuscript is too long. Specifically, the introduction and discussion should be abbreviated to include only the most pertinent information. Most of the background information should be made more concise. In general, any claim should be supported by a relevant citation. The first several paragraphs of the introduction have many claims without citation and are quite verbose. Recommend making the introduction much more concise.

>>We have edited the manuscript to shorten it and remove unnecessary information. 

2. A full discussion of the subject inclusion and exclusion is recommended. This should include both demographic inclusion criteria as well as clinical. Furthermore, the recruitment strategy should be discussed. Also, if any compensation was involved. Please describe the consenting process and other ethical considerations, including the parental roles.

>> We have included additional information regarding recruitment processes, the consent process, parental/guardian involvement and the compensation. 

Added text

"Methods

Participants

...Participants in the POND Network who agreed to receive study recruitment emails received a study flyer via email and were invited to contact the study’s research coordinator for more information if interested. 

...Participants were given a description of the study and their understanding of the study was assessed through a series of questions. Upon successfully demonstrating their understanding of the study, and if they were still willing to participate, participants and their parent or guardian signed the consent form. As a token of appreciation for their participation, participants were given a gift card (regardless of successful study completion). A parents or guardian was present during the consent process, but not during the experimental protocol." 

3. This reviewer questions the decision to include the “excluded” subjects in the analysis. Including these subjects, and also analyzing the cohort without them leads to a confusing description of the results. Furthermore, table 2 is quite confusion with these two groups described (with and without usable NIR data). Recommend only reporting results on subjects with a complete dataset.

>>We have modified the statistical results presented to only report on subjects with a complete dataset. Table 2 has also been updated.

Added text:

"Results

Results are presented for the remaining 10 participants in the ASD group. 

Participant characteristics

The two participant groups were not significantly different with respect to age (p = 0.76, t16 = -0.309), but differed significantly with respect to IQ ( p = 0.01, t15 = -3.23). 

Subjective pain ratings

The pain ratings were significantly different between the two task types (p < 0.01, F70 = 241.663) but did not different significantly between the two study groups (p = 0.628, F14 = 0.245). 

Pain/discomfort tolerance

Statistical analysis of the cold-water trial durations showed no significant differences were observed between the two groups or across the three trials (group: p = 0.09, F45 = 7.483; trial: p = 0.406, F31 = 0.929)." 

4. Table 2 should also include p values.

>> Table 2 has been updated to include p-values.

Please see the attached "Response to Reviewers" document for a formatted version of table 2. 

5. The results section should report on the data only. Further description of methods should be placed in the “methods” section, including the section starting on page 15 line 354

>>The manuscript has been edited such that the results section only includes data, and the methods have been moved to the methods section.

Added text:

"Feature extraction

Because participants dictated the duration of the cold stimulus trials, the length of the trials varied across participants and task repetition. All control trials were 60s long, while the stimulus trials were up to 180s. To compare the evolution of the hemodynamic response between trial types, trial duration ought to be consistent. Thus, two different comparisons of brain activity were considered for statistical analysis of functional brain measurements: i) maximum [HbO] across the full task intervals and ii) maximum [HbO] within the first 60s of the task intervals."

---

## [Decision Letter · Decision Letter 1]

14 Jun 2021

PONE-D-21-06009R1

Investigating Sensory Response to Physical Discomfort in Children with Autism Spectrum Disorder using Near-Infrared Spectroscopy

PLOS ONE

Dear Dr. Doyle-Thomas,

Thank you for submitting your manuscript to PLOS ONE. After careful consideration, we feel that it has merit but does not fully meet PLOS ONE’s publication criteria as it currently stands. Therefore, we invite you to submit a revised version of the manuscript that addresses the points raised during the review process.

Two experts in the field, one is the same, have carefully reviewed the revision. The original #1 reviewer is satisfied with the revised manuscript, while the newly participated reviewer #3 is almost satisfied with the revision leaving some minor concerns which should be clarified before publication.  

I am looking forward receiving the necessary revision and the replies to the critiques. 

We look forward to receiving your revised manuscript.

Kind regards,

Manabu Sakakibara, Ph.D.

Academic Editor

PLOS ONE

Journal Requirements:

Reviewers' comments:

Reviewer's Responses to Questions

**Comments to the Author**

1. If the authors have adequately addressed your comments raised in a previous round of review and you feel that this manuscript is now acceptable for publication, you may indicate that here to bypass the “Comments to the Author” section, enter your conflict of interest statement in the “Confidential to Editor” section, and submit your "Accept" recommendation.

Reviewer #1: All comments have been addressed

Reviewer #3: All comments have been addressed

2. Is the manuscript technically sound, and do the data support the conclusions?

Reviewer #1: (No Response)

Reviewer #3: Yes

3. Has the statistical analysis been performed appropriately and rigorously? 

Reviewer #1: (No Response)

Reviewer #3: Yes

4. Have the authors made all data underlying the findings in their manuscript fully available?

Reviewer #1: (No Response)

Reviewer #3: Yes

5. Is the manuscript presented in an intelligible fashion and written in standard English?

Reviewer #1: (No Response)

Reviewer #3: Yes

6. Review Comments to the Author

Reviewer #1: (No Response)

Reviewer #3: The authors explored the objective pain assessment tools in ASD by using NIRS and achieved some results. This study was well written after revision. There were some concerns about the methods as follow:

1. The effects of using psychotropic drugs and cognitive level in ASD children should be considered in this study. The IQ level differed significantly between 2 groups, so the authors should discuss the influence of IQ level as a confounding factor.

2. The authors expressed that self-reporting of pain can be difficult in ASD children and "no history of chronic pain" was one of the inclusion/exclusion criteria. How to confirm "no history of chronic pain" in ASD children？

7. PLOS authors have the option to publish the peer review history of their article (what does this mean?). If published, this will include your full peer review and any attached files.

Reviewer #1: **Yes: **Rhonda Manning, PT, DPT

Reviewer #3: No

---

## [Author Response · Author response to Decision Letter 1]

23 Jul 2021

Response - Investigating Sensory Response to Physical Discomfort in Children with Autism Spectrum Disorder using Near-Infrared Spectroscopy

Reviewer #3: The authors explored the objective pain assessment tools in ASD by using NIRS and achieved some results. This study was well written after revision. There were some concerns about the methods as follow:

1. The effects of using psychotropic drugs and cognitive level in ASD children should be considered in this study. The IQ level differed significantly between 2 groups, so the authors should discuss the influence of IQ level as a confounding factor.

Regarding the potential effect of medication on the hemodynamic response, this limitation was already addressing in the Limitation section on page 21, Ln 496. Regarding IQ, while the two groups differed significantly in IQ, the task did not have a significant cognitive component. As addressed in section Sample Size, previous work has shown that cognitive level does not significantly affect sensory reactivity. The text has been modified to clarify this. 

Added text:

Sample Size

Additionally, study groups were not matched for IQ in this work. Although this is a potential confounding factor, it has been found that cognitive level (IQ) or overall developmental level is not related to abnormal sensory reactivity in children with ASD2.

2. The authors expressed that self-reporting of pain can be difficult in ASD children and "no history of chronic pain" was one of the inclusion/exclusion criteria. How to confirm "no history of chronic pain" in ASD children？

“No chronic history of pain” was confirmed by the participant and their caregiver during the consent process. 

Added text:

Method

Participants

It was also confirmed that participants met the inclusion/exclusion criterion during the consent process and were eligible to participate. Upon successfully demonstrating their understanding of the study and confirming eligibly criterion were met, and if they were still willing to participate, participants and their parent or guardian signed the consent form.

---

## [Decision Letter · Decision Letter 2]

23 Aug 2021

Investigating Sensory Response to Physical Discomfort in Children with Autism Spectrum Disorder using Near-Infrared Spectroscopy

PONE-D-21-06009R2

Dear Dr. Doyle-Thomas,

We’re pleased to inform you that your manuscript has been judged scientifically suitable for publication and will be formally accepted for publication once it meets all outstanding technical requirements.

Kind regards,

Manabu Sakakibara, Ph.D.

Academic Editor

PLOS ONE

Additional Editor Comments (optional):

Reviewers' comments:

Reviewer's Responses to Questions

**Comments to the Author**

1. If the authors have adequately addressed your comments raised in a previous round of review and you feel that this manuscript is now acceptable for publication, you may indicate that here to bypass the “Comments to the Author” section, enter your conflict of interest statement in the “Confidential to Editor” section, and submit your "Accept" recommendation.

Reviewer #1: All comments have been addressed

Reviewer #3: All comments have been addressed

2. Is the manuscript technically sound, and do the data support the conclusions?

Reviewer #1: Yes

Reviewer #3: Yes

3. Has the statistical analysis been performed appropriately and rigorously? 

Reviewer #1: Yes

Reviewer #3: Yes

4. Have the authors made all data underlying the findings in their manuscript fully available?

Reviewer #1: Yes

Reviewer #3: Yes

5. Is the manuscript presented in an intelligible fashion and written in standard English?

Reviewer #1: Yes

Reviewer #3: Yes

6. Review Comments to the Author

Reviewer #1: Study manuscript is significantly improved. A possible explanation for the hemispheric differences note may be the altered connectivity of the brains in children with ASD. Decreased inter-hemispheric connectivity with increased intra-hemispheric connections may account for this altered pattern.

The data presented in the tables and figures is easy to follow and read. Well done!

Reviewer #3: (No Response)

7. PLOS authors have the option to publish the peer review history of their article (what does this mean?). If published, this will include your full peer review and any attached files.

Reviewer #1: **Yes: **Rhonda Manning, PT, DPT, PCS

Reviewer #3: No

---

## [Editor Report · Acceptance letter]

26 Aug 2021

PONE-D-21-06009R2 

Investigating Sensory Response to Physical Discomfort in Children with Autism Spectrum Disorder using Near-Infrared Spectroscop 

Dear Dr. Doyle-Thomas:

I'm pleased to inform you that your manuscript has been deemed suitable for publication in PLOS ONE. Congratulations! Your manuscript is now with our production department. 

Kind regards, 

on behalf of

Dr. Manabu Sakakibara 

Academic Editor

PLOS ONE